# Laparoscopic Evaluation of the Reproductive Tract in Two Female Polar Bears (Three Procedures) (*Ursus maritimus*)

**DOI:** 10.3390/life14010105

**Published:** 2024-01-09

**Authors:** Ellison D. Aldrich, Dean A. Hendrickson, Todd L. Schmitt, Hendrik H. Nollens, Gisele Montano, Karen J. Steinman, Justine K. O’Brien, Todd R. Robeck

**Affiliations:** 1School of Veterinary Sciences, Massy University, Palmerston North 4442, New Zealand; 2College of Veterinary Medicine and Biomedical Sciences, Colorado State University, Fort Collins, CO 80521, USA; 3SeaWorld of California, 500 Sea World Drive, San Diego, CA 92109, USA; 4Sand Diego Zoo Wildlife Alliance, 15500 San Pasqual Valley Road, Escondido, CA 92027, USA; 5Species Preservation Laboratory, SeaWorld Parks and Entertainment Corporation, 2595 Ingraham Road, San Diego, CA 92109, USAtodd.robeck@seaworld.com (T.R.R.); 6Taronga Conservation Society Australia, Mosman, NSW 2088, Australia

**Keywords:** polar bear, laparoscopy, ovary, ultrasound, reproduction

## Abstract

Polar bears (*Ursus maritimus*) face a number of challenges that threaten the survival of the species. Captive breeding represents one essential facet of species conservation, but aspects of the polar bear’s reproductive physiology, such as follicle maturation, coitus-induced ovulation, and pseudopregnancy, are poorly characterized and present challenges for enhancing natural reproductive success and the application of advanced reproductive techniques. Due to the absence of a reliable transrectal or transabdominal ultrasound method for ovarian examination in the species, the ovaries of two adult female polar bears were examined laparoscopically to evaluate the feasibility of surgical access to the ovaries, oviduct, and uterus. The minimally invasive procedure was easily and rapidly performed in both bears and all procedures. Direct visual assessment of the ovary was possible after dissection of a fatty bursal sac, which completely enclosed the ovaries. In the second bear, laparoscopic manipulation of the ovary to draw it closer to the body wall enabled transcutaneous ultrasound. Laparoscopy may be a valuable tool to aid in the application of advanced reproductive technologies in polar bears.

## 1. Introduction

Polar bears (*Ursus maritimus*) face a number of challenges that threaten the survival of the species, including climate change, which results in altered food availability and loss of sea ice habitat [1,2], as well as xenoendocrine pollutants [3]. In addition to broader conservation efforts, captive breeding programs will likely have a vital role in the preservation of the species. To improve the success of captive breeding, it is essential to gain a greater understanding of the reproductive anatomy and physiology of the polar bear in order to implement practical and effective strategies for reproductive management.

It has been demonstrated that many bear species, including the polar bear, are seasonally polyestrous and considered to be induced ovulators [4,5,6,7,8]. The breeding season in polar bears is typically between February and April. After conception, the blastocyst enters diapause for five to six months, after which implantation occurs [9]. Following a ~60-day gestation, cubs are born in November or December [6,10,11]. If conception does not occur at the time of ovulation, pseudopregnancy may occur, which is characterized by an increase in progestagens at the normal time of implantation. Both pseudopregnancy and pregnancy result in increased progestagen concentrations, making it difficult to distinguish a true pregnancy from pseudopregnancy, failure of implantation after diapause, fetal resorption, or abortion [12]. This complex reproductive physiology presents a distinct set of challenges for the application of assisted reproductive technologies (ARTs) such as artificial insemination (AI) and complicates reliable pregnancy detection through the monitoring of serum, urine, and fecal progestogen metabolites. Ovarian stimulation and ovulation induction with exogenous gonadotropins followed by AI has been described in one polar bear, although no cubs were produced from that attempt [6]. Visualization of the ovaries using transrectal or transabdominal ultrasound would provide information on follicular maturation and ovulation timing, both of which are critical to the success of AI. Attempts by our group to visualize polar bear ovaries’ ultrasonographically using various probes (4–10 mHz) and probe extensions (probe inserted until encountering the kidney) in lateral recumbency have been unsuccessful, and to our knowledge, no published reports of a validated repeatable method exist. In addition, due to the intractable nature of the animals, even non-invasive transabdominal ultrasonographic examinations will in most circumstances require general anesthesia.

In many other domestic species, ARTs are utilized frequently. Laparoscopic insemination is well established in sheep and goats [13,14]. In the standing mare, laparoscopic evaluation of oviductal patency and oviductal flushing with PGE_2_ or topical application of PGE_2_ gel has been reported [13,15,16,17], and laparoscopic ovariectomy is now routine [18]. Minimally invasive surgical instrumentation is readily available for domesticated animals. The instrumentation used in the horse can be easily adapted to bears.

Laparoscopy has been reported in the North American black bear and Asiatic black bear [5,19,20], but to our knowledge, this report is the first account of laparoscopy in the polar bear. The anatomy of the polar bear ovary is not well described in the literature. The objective of this study was to describe the anatomy and ultrasound appearance of the polar bear ovary and to describe the laparoscopic approach to the ovaries.

## 2. Materials and Methods (Case Report)

All procedures described herein were reviewed and approved by the SeaWorld Parks and Entertainment Animal Research and Welfare Committee and were performed in accordance with the U.S Animal Welfare Act. Two nulliparous, captive-born adult polar bears aged 20.5 years, were included in this project. For each procedure, each polar bear (bear 1 (Studbook #1051) one procedure and bear 2 (Studbook #1098) two procedures, 1 year apart, for a total of three procedures) was darted in the shoulder region using a TeleDart pistol and 5 mL Teledart with a 2 mm × 40 mm collared needle (TeleDart USA, Boulder, CO, USA) to administer a combination of compounded 40 mg/mL medetomidine (ZooPharm LLC, Laramie, WY, USA 0.05 mg/kg), compounded 200 mg/mL ketamine (ZooPharm LLC, Laramie, WY, USA, 3.3 mg/kg), and compounded 50 mg/mL midazolam (ZooPharm LLC, Laramie, WY, USA, 0.19 mg/kg). This protocol produced a consistent rapid onset of immobilization that was partially reversible during anesthesia via atipamezole (Zoetis Inc., Kalamazoo, MI, USA, 0.25 mg/kg) and flumazenil (West-Ward, Eatontown, NJ, USA, 0.05 mg/kg). Once immobilization was complete and the bear appeared safe to approach, a head restraint was used to safeguard the head and evaluate the reflexes of the bears. For each procedure, each female was intubated with a 20 mm endotracheal tube and placed on gas anesthesia with sevoflurane (SevoFlo^®^, Abbott Laboratories North Chicago, IL, USA) at 8% initially. Assisted ventilation with inspiratory peak pressure < 25 cm H_2_O was added to improve ventilation and perfusion while the bears were positioned in a recumbency between lateral and dorsal during the laparoscopy. Sevoflurane was maintained at 3% during the procedure. Bears received fluid therapy during the procedure with an IV catheter in the jugular or cephalic vein, intraoperative IV antibiotics (ceftazidime, Sagent, Pharmaceuticals, Schaumburg, IL, USA; 20 mg/kg) and an intramuscular nonsteroidal anti-inflammatory (meloxicam, Covetrus North America, Dublin, OH, USA; 0.15 mg/kg).

For the procedure, bears were positioned in recumbency 45° between right lateral and dorsal recumbency with the upper limbs suspended from vertical poles fixed to the surgical table (Figure 1A,B). An approximately 20 × 40 cm rectangular area of fur was clipped from the abdomen, centered at the umbilicus. The area was then aseptically prepared and draped.

Laparoscopic evaluation was performed similarly in both bears, beginning with the creation of a 15 mm incision at the caudal aspect of the umbilicus through the skin, subcutaneous tissue, and the linea alba. An 11 mm diameter, 20 cm long cannula with a blunt obturator (Surgical Direct, Deland, FL, USA) was advanced into the peritoneal space. A 10 mm diameter, 46cm long, 30° forward-viewing telescope (WA52005A, Olympus Winter & Ibe GmbH, Hamburg, Germany) was placed through the cannula to confirm location within the abdomen. After confirmation of correct placement, the abdomen was insufflated to an intra-abdominal pressure of 12 mmHg with CO_2_ at a rate of 12 L/min. The left side of the abdomen was evaluated. Two other portals (11 mm diameter, 20 cm long cannula with a blunt obturator) were placed under intra-abdominal observation in the first bear, 10 and 15 cm caudal to the umbilical portal, and one portal 10 cm caudal to the umbilical portal in the second bear (two procedures). The bladder was located and was noted to be partially full of urine and air secondarily to endoscopically guided catheterization of the uterus for AI, which was performed concurrently. The left uterine horn was identified and traced to the oviduct and then ovary using 10 mm diameter, 45 cm long Babcock grasping forceps (Surgical Direct, Deland, FL, USA) (Figure 2A). The ovary was located ~4 cm caudal to the kidney and was completely covered by a fibrous and fatty bursa (Figure 2B). The surface of the ovary was not visible through the bursal sac. The bursa was carefully evaluated, and no pre-existing anatomical opening to the ovary was observed. In bear 1, the fatty covering of the left ovary was bluntly dissected using Babcock forceps or laparoscopic scissors to reveal the ovary (Figure 2C). The left ovary was noted to have two follicles, one approximately 0.5 cm in diameter and one approximately 0.3 cm in diameter. The bursa was placed back over the ovary. The right uterine horn, oviduct, and ovary were identified and visually evaluated in situ without grasping. The location of the second ovary was identified, and transabdominal ultrasonography was attempted using a 3–5 mHz curvilinear probe (GE Logiq Book, GE Medical Systems, Milwaukee, WI, USA); however, the abdominal air interfered with the quality of the image. In response, the abdomen was desufflated, and a second attempt was made to evaluate the ovary transabdominally again without examining the right ovary because of the unknown consequences of penetrating the fatty bursa. In the second and third procedures, performed on bear 2, the ovary was grasped with Babcock forceps to manipulate it for visual evaluation and then held adjacent to the body wall (Figure 3A,B). Ovarian ultrasound was performed transabdominally using a 3–5 mHz curvilinear probe (GE LOGIQ e Vet, GE Healthcare, Milwaukee, WI, USA). In bear 2, during the first procedure, one follicle was seen in each ovary ultrasonographically measuring 1.78 × 1.83 cm in diameter (left ovary, Figure 3C) and 1.21 × 1.28 cm (right ovary). Correct positioning of the ultrasound probe was confirmed by gentle ballottement of the ovary (by the hand holding the probe). Laparoscopic video and ultrasound images were recorded. The ultrasound was facilitated by reducing the pneumoperitoneum as the ovary was held to the body wall. The left ovary was moved to the left body wall and the right ovary to the right inguinal area during the first laparoscopy in bear 2 and the left body wall in the second laparoscopy. The ability to elevate the right ovary to the left body was accomplished by first desufflating the abdomen. This technique made it easier to visualize both ovaries via ultrasound from the same transabdominal location and minimized potential contamination of the surgical site. Ovaries were located at ~30 cm cranial to the vulva (Figure 3B). In the second procedure on bear 2, both ovaries were elevated to the left flank for ultrasound evaluation. The left ovary had a structure indicative of a corpus hemorrhagicum (1.13 × 1.08 cm) and two follicles (F1: 1.53 × 1.53 cm; F2: 0.49 × 0.54 cm), and the right ovary had two small follicles (0.3 to 0.5 cm in diameter). The polar bears in each of these procedures had been hormonally induced, which likely led to the ability to find the follicles and corpus hemorrhagicum. It may be more difficult to find these structures within the ovary without hormonal induction.

In both bears and all three procedures, the body wall incisions were closed with size 0 polyglyconate suture (Maxon™ Covidien™ LLC, 15 Hampshire St. Mansfield, MA, USA) in a simple interrupted pattern. The skin was closed with 2-0 polyglyconate suture in a simple continuous intradermal pattern. Cyanoacrylate glue was then used to seal the incisions. The operative portion of the procedure averaged 15 min.

No intra-operative complication occurred, and both bears recovered well from general anesthesia. However, bear 2 showed minor discomfort with an elevated respiratory rate and reduced activity that lasted for six hours after the first procedure. Two to four hours after the procedure, they were offered water, and normal feeding resumed within four to six hours. For pain management, both bears received meloxicam (Covetrus North America, Dublin, OH, USA; 0.16 mg/kg PO) intraoperatively and then s.i.d. for 5 additional days. Both bears had normal appetite and behavior the next day post-surgery and no additional signs of pain were observed.

## 3. Discussion

In both bears, laparoscopy via a telescope portal at the caudal aspect of the umbilicus provided access for in situ visual assessment of both ovaries, oviducts, and uterine horns. The positioning of the bear at 45° from dorsal recumbency provided easy access to both uterine horns, oviducts, and ovaries. Liehmann et al. showed that tilting dogs to 45° from dorsal recumbency provided the best observation of the ovaries in dogs [21]. Although dorsal recumbency is often used to begin laparoscopic procedures, it was determined that the size of the animal would make it more difficult to change the positioning during the surgical procedure without increasing the risk of possible contamination of the surgical site. During the surgical procedure, it was relatively easy to access the majority of the reproductive tract from this position. It did not seem necessary to move the animal from one recumbency to another. The long uterine horns, long broad ligament, and relatively long mesovaria contributed to being able to manipulate the “down” ovary to the body wall. Two additional instrument portals were necessary to dissect the ovary free from the fatty bursa that encases the ovaries in the polar bear, whereas only a single additional portal was required to elevate the ovaries for ultrasound evaluation. The reproductive anatomy of the female polar bear is not well described in the literature. Based on reports in other bear species, the bursa completely surrounding the ovaries was an unexpected finding. Boone et al. utilized laparoscopy to monitor the development of corpora lutea post-mating in black bears [5]. In the black bear, the ovary was surrounded by a thin membrane of tissue and the corpora lutea were easily identified. In contrast, our findings in the polar bear indicate that this would not be possible due to the thick and opaque nature of the fat impregnated bursal sac entirely surrounding the ovaries. It is possible that the body condition of the polar bears in this setting would be different than that of polar bears in a wild setting. The “good” or “over” condition of captive polar bears may make it more difficult to successfully ultrasound ovarian structures than would be encountered in wild polar bears that are “less” conditioned.

Manipulation of the ovary in bear 2 so that the ovary was maintained directly against the body wall in the inguinal area facilitated transabdominal ultrasound and evaluation of follicular development. It is important to note that it is essential to clip the inguinal area and include the region in the aseptic prep. Without laparoscopic assistance, the ovary could not be successfully located by transabdominal nor transrectal ultrasound in the polar during previous procedures by our group. In retrospect, our inability to locate the ovary during these procedures is believed to have been due in part to the thick fatty bursal sac combined with the subcutaneous fat, tissues which are known to attenuate ultrasonographic images. If the goal is to evaluate follicular size the laparoscopic manipulation and transcutaneous ultrasound reported herein obviate the need to penetrate the bursal sac surrounding the ovary and provide a mechanism to evaluate the ovarian structures quickly and reliably. This method would seem preferred to bursal dissection, but because the function of the bursa is unknown, there is no way to predict the effects of surgical dissection on immediate post-operative fertility. Alternatively, the use of a laparoscopic ultrasound probe may be a viable option for evaluating follicular development and the presence of corpora lutea. Laparoscopic ultrasound has been described in dogs [22]. As mentioned previously, there are no reports of ovarian ultrasound imaging in the polar bear, and it is likely that the relatively small ovarian size, the fatty bursa that surrounds the ovary, and the “fat like” echogenicity of the ovaries make them very difficult to identify in the large polar bear abdomen.

During the laparoscopic procedure in this report, intrauterine AI was performed simultaneously using a flexible endoscope inserted transvaginally, but neither of the bears produced cubs, despite one bear (bear 2) displaying elevated progestagens during the expected period of embryo implantation and placental gestation for the species after the procedure (details to be covered in a forthcoming publication).

Laparoscopic surgery is increasingly prevalent in both domestic and zoo species due to the minute size of incisions, minimally invasive nature of the procedure, and minimal post-operative morbidity. In captive wild animals in particular, there are extreme limitations to the feasibility of managing incisional complications such as infection or dehiscence; therefore, a shift away from traditional open approaches whenever possible is logical. In many instances, laparoscopic surgery may offer superior visibility and access to the abdominal cavity. In the most recent report of laparoscopy in bears, Pizzi et al. described laparoscopic cholecystectomy in nine Asiatic black bears [19]. Their technique was similar with regard to the location of the telescope portal; however, bears were positioned in dorsal recumbency, surgeons operated from a position between the hind legs, and instrument portals were located cranial in order to access the liver and gall bladder. All bears in the current study had rapid recoveries and resumed normal behavior within 12 h of surgery. The post-laparoscopic discomfort observed in bear 2 (procedure 1) may have been the result of incomplete evacuation of insufflation gas prior to closure. As mentioned previously, to safely perform either rectal or transabdominal ultrasonography in a polar bear, the bear must be anesthetized, and therefore, the minimally invasive nature of laparoscopy adds very few potential complications to the procedure.

## 4. Conclusions

Laparoscopy is a viable technique for accessing the ovaries in the polar bear and can be used to assist in transcutaneous ultrasound or to perform direct manipulation of the reproductive tract. To successfully develop assisted reproductive techniques in this and any species, it is crucial to validate the ovulatory response to induction protocols. In addition, for the polar bear, it is also vital to determine if intrauterine or oviductal insemination via laparoscopy are tools that could be relied on in the future. Studies are needed to further characterize the temporal relationships among hormone patterns and ovulation in females undergoing natural breeding and exogenous ovarian stimulation and ovulation induction to ensure optimal timing of insemination and successfully apply advanced reproductive techniques. These results provide evidence that laparoscopy may be a valuable tool to aid in this process.

## Figures and Tables

**Figure 1 life-14-00105-f001:**
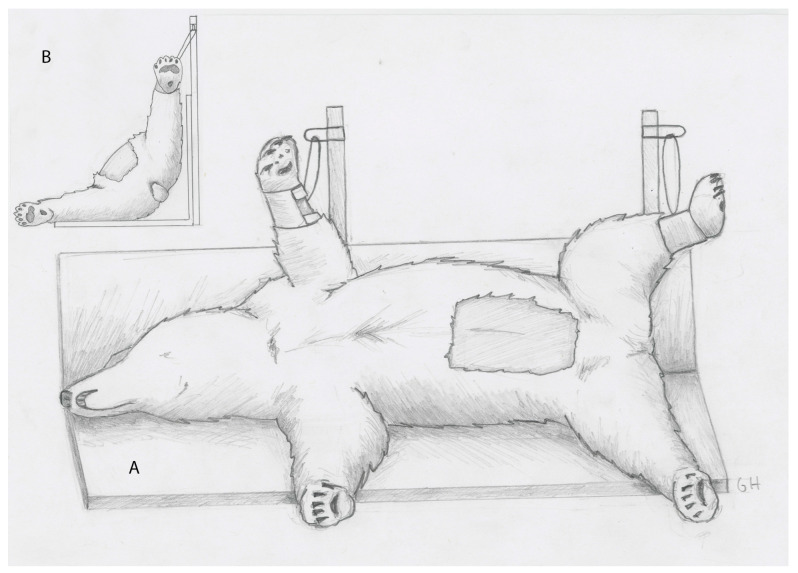
(**A**) Drawing showing the positioning of the polar bear halfway between dorsal and lateral recumbency. The shaded area over the abdomen represents the area clipped for surgery. (**B**) Drawing showing the positioning of the bear from the rear angle.

**Figure 2 life-14-00105-f002:**
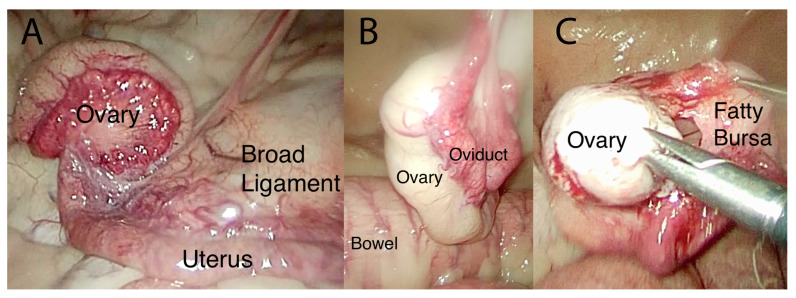
(**A**). Laparoscopic photograph of the left ovary, uterine horn, and broad ligament in the normal position. (**B**). Laparoscopic photograph of the left ovary (covered by the fatty bursa), oviduct, and bowel. (**C**). Laparoscopic photograph of the same ovary with the fatty bursa pulled away after blunt dissection.

**Figure 3 life-14-00105-f003:**
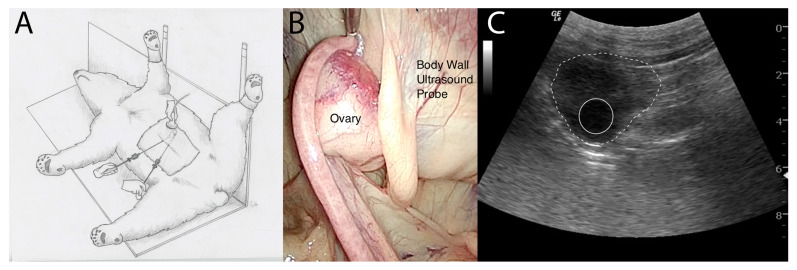
(**A**). Drawing of the bear showing the positioning of the instruments and ultrasound probe. (**B**). Laparoscopic photograph of the left ovary held adjacent to the body wall for ultrasound examination. (**C**). Ultrasound image of left ovary of bear 2 during the first surgical procedure. The solid white circle outlines a 1.8 mm follicle, and the dashed white line outlines the left ovary.

## Data Availability

Data are contained within the article.

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
