# Peer review of "Laparoscopic Evaluation of the Reproductive Tract in Two Female Polar Bears (Three Procedures) (Ursus maritimus)"

_life, 2024, doi:10.3390/life14010105_

Round 1

Reviewer 1 Report

Comments and Suggestions for Authors

See the comments made in the pdf.

Comments on the Quality of English Language

Minor error mentioned in one of the comments.

Author Response

Thank you for the thoughtful review and suggested changes. Here are my responses:

Line 3: the spelling of procedures has been corrected.

Line 68: We have added a sentence re objectives.

Line 124 and 130: We do not have a laparoscopic image of the follicle, just an ultrasound image that is included in Figure 3C

Line 174: We removed the redundant verbiage and moved the second part of the sentence to follow the first sentence of the paragraph.

Line 252: We added "is" to this sentence.

Reviewer 2 Report

Comments and Suggestions for Authors

Diagnostics and therapy of wild animals is a very important but extremely important branch of veterinary practice. The use of specialized knowledge and unique practical skills are extremely important in the protection of rare wild animals, including species threatened with extinction. Wild animal disease specialists are extremely valuable to zoos and national parks. They also serve as field practice consultants. The authors of the manuscript undertook a detailed description of the laparoscopic examination of the reproductive tract in a female polar bear. This is extremely important due to the specific physiology and reproductive biology of this animal. I congratulate the authors on a professionally prepared article, which is also written in a very accessible way. I am convinced that it will be highly appreciated by readers, both scientists and practitioners.

Author Response

Thank you for your input. I do not see any comments to change.

Reviewer 3 Report

Comments and Suggestions for Authors

In my opinion the manuscript is well written. All procedures were described in details, thus they are repeatable. The results of this case report have clinical importance, especially for veterinarians working in zoological gardens. 

My minor suggestion to the Authors:

- the position of the reference citations should be corrected - please check the author guidelines

Author Response

Thank you for your review. We will complete a review of the citation position.